# *NTPP*: Generative Speech Language Modeling for Dual-Channel Spoken Dialogue via Next-Token-Pair Prediction

Qichao Wang [* 1]  Ziqiao Meng [* 1 2]  Wenqian Cui [3]  Yifei Zhang [4]  Pengcheng Wu [4]  Bingzhe Wu [5]  Irwin King [3]
Liang Chen  Peilin Zhao [1 6]

## Abstract

Inspired by the impressive capabilities of GPT-4o, there is growing interest in enabling speech language models (SLMs) to engage in natural, fluid spoken interactions with humans. Recent advancements have led to the development of several SLMs that demonstrate promising results in this area. However, current approaches have yet to fully exploit dual-channel speech data, which inherently captures the structure and dynamics of human conversation. In this work, we systematically explore the use of dual-channel speech data in the context of modern large language models, and introduce a novel generative modeling paradigm—***Next-Token-Pair Prediction (NTPP)***—to enable *speaker-independent* dual-channel spoken dialogue learning using *decoder-only* architectures for the first time. We evaluate our approach on standard benchmarks, and empirical results show that our proposed method, NTPP, significantly improves the conversational abilities of SLMs in terms of turn-taking prediction, response coherence, and naturalness. Moreover, compared to existing methods, NTPP achieves substantially lower inference latency, highlighting its practical efficiency for real-time applications. Demo and code can be found at https://audio-3059.pages.dev.

---

*Equal contribution. This work is done when the first two authors work as interns in Tencent AI Lab. [1]Tencent [2]National University of Singapore [3]The Chinese University of Hong Kong [4]Nanyang Technological University [5]Shenzhen University [6]Shanghai Jiao Tong University. Correspondence to: Ziqiao Meng <zq-meng@nus.edu.sg>, Peilin Zhao <masonzhao@tencent.com>.

*Proceedings of the 42nd International Conference on Machine Learning*, Vancouver, Canada. PMLR 267, 2025. Copyright 2025 by the author(s).

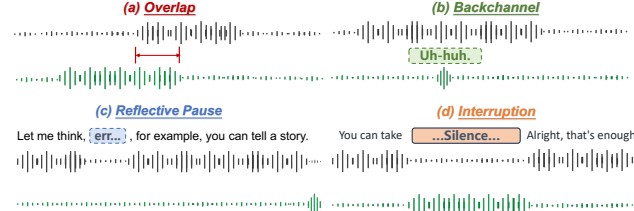

*Figure 1.* The dual-channel speech encapsulates various conversational turn-taking events, including: (a) Overlap, (b) Backchannel, (c) Pause, and (d) Interruption. These events are intermingled within the single-channel audio stream but could be explicitly observed in the dual-channel audio stream.

## 1. Introduction

The emergence of large language models (LLMs), especially the GPT series as referenced in (Patel et al., 2023; OpenAI, 2023; 2024), has significantly revolutionized the realm of artificial intelligence. These potent language models (LMs) derive their capabilities from pretraining on vast text corpora, utilizing decoder-only transformer architectures, and are steered by a ***next-token prediction (NTP)*** objective function. Recently, there's been a surge of interest in merging LLMs with other modalities, such as images (Radford et al., 2021; Li et al., 2023; Liu et al., 2023b), audio (Zhang et al., 2023a; Hassid et al., 2023), protein sequences (Lin et al., 2022; Madani et al., 2023), and more. Among these modalities, audio or speech data is particularly crucial as it allows LLMs to engage in real-time vocal interactions with humans. The recently introduced GPT-4o model (OpenAI, 2024) demonstrates exceptional proficiency in handling real-time interactions with users in conversational contexts. During the demo presentation, it was capable of generating genuine emotional responses and engaging users with prompt reactions. However, these functionalities present additional challenges, as the model must accurately interpret the unique audio information embedded in human speech while performing inference with minimal delay.

A wide range of advanced speech language models (SLMs) (Xie & Wu, 2024; Zhang et al., 2023a; Fang et al., 2024; Hassid et al., 2023; Rubenstein et al., 2023; Nguyen et al.,

*Table 1.* Comparisons to existing dual-channel spoken dialogue generative models.

| Model | Speaker-Independent | Encoder-Free | VAD-Free | Single KVCache |
|---|---|---|---|---|
| dGSLM | ✓ | | ✓ | |
| LSLM | | ✓ | | ✓ |
| Moshi | | | ✓ | |
| **NTPP (Ours)** | ✓ | ✓ | ✓ | ✓ |

2024; Fathullah et al., 2024; Nachmani et al., 2024) has been developed to enable real-time voice interactions with human users. These models typically rely on single-channel audio data and some of them focus on aligning audio and text streams. However, the potential of dual-channel speech data has been somewhat under-explored. Dual-channel speech, which records the audio channels of two speakers independently, offers distinct advantages over single-channel data. Notably, it can explicitly capture various conversational dynamics, such as overlaps, pauses, and interruptions, providing a richer representation of real-world interactions. Some examples of these turn-taking events are illustrated in Figure 1. These dynamics can help train SLMs to engage in more natural and fluent conversations with human users across diverse scenarios.

dGSLM (Nguyen et al., 2023) was the first to introduce textless generative spoken dialogue modeling for simulating dual-channel speech using a Siamese two-tower transformer architecture. More recently, LSLM (Ma et al., 2024) proposed token fusion strategies, in which dual-channel audio tokens are combined and fed into a causal transformer. Moshi (Défossez et al., 2024), in contrast, presents a text-based multi-channel speech sequence model that aligns multi-scale audio streams with textual streams in parallel. However, these approaches generally either rely on an additional encoder—randomly selecting one speaker's channel as the input condition—or lack speaker-independence, meaning the learned distribution is not permutation invariant with respect to speaker order.

In this research, we propose a novel dual-channel speech autoregressive generative model based on an innovative paradigm called ***next-token-pair prediction (NTPP)***, utilizing a decoder-only transformer architecture. The model is trained to predict the next pair of speech tokens conditioned on previously generated spoken dialogues. This approach capitalizes on the time-aligned structure of dual-channel speech sequences, enabling a more precise representation of the generative distribution in spoken dialogues. To expand its applicability in advanced speech language models, we extend our solution from vector quantization (VQ) tokenizers (van den Oord et al., 2017) to the more advanced residual vector quantization (RVQ) (Lee et al., 2022) tokenizers.

Compared to existing approaches, NTPP offers four key advantages. First, instead of modeling a conditional distribution, NTPP directly estimates the joint distribution of both speakers. Second, it adopts a decoder-only architecture, which provides improved learning and parameter efficiency compared to models requiring additional encoders (Dao et al., 2022). Third, NTPP eliminates the need for a voice activity detection (VAD) module, learning diverse turn-taking behaviors in a fully data-driven manner. Fourth, it maintains a single KVCache (Pope et al., 2023b), enabling greater memory and inference efficiency. Table 1 summarizes these advantages in comparison to existing methods.

We conduct comprehensive experiments to evaluate the performance of our approach across multiple dimensions. First, we assess continual dialogue generation by providing preceding conversational context, following established benchmarks (Nguyen et al., 2023). We further evaluate response success rates during interruption events in synthesized streaming multi-turn conversations. The results demonstrate that our method enables SLMs to more effectively model the distribution of turn-taking events in spoken dialogue. In addition, we perform human evaluations to assess the meaningfulness and naturalness of the generated responses. To test speaker independence, we permute the input channels of different speakers and observe the robustness of model performance; our NTPP exhibits the highest stability among all baselines. Finally, we measure inference latency to determine whether the model can generate timely and coherent responses—excluding initial non-informative tokens. Our findings show that NTPP delivers faster response times, with performance remaining strong, particularly as the number of conversation turns increases. To sum up, our contributions can be summarized as follows:

- We introduce a novel next-token-pair prediction (NTPP) paradigm for generative spoken dialogue modeling, implementing a decoder-only architecture with innovative design enhancements.

- We develop compatible solutions for both VQ and RVQ speech tokenizers, enabling a broad range of SLMs to effectively learn from dual-channel speech using our proposed method.

- We comprehensively evaluate NTPP, highlighting its strengths in conversational event simulation, speaker independence, and inference latency, among others.

## 2. Related Works

**Speech Language Models (SLMs).** Recent advancements in SLMs have focused on integrating LLMs (OpenAI, 2023; Dubey et al., 2024; Chu et al., 2024) to unify audio and text processing (Ao et al., 2022; Tang et al., 2022; Wang et al., 2023; Rubenstein et al., 2023). Some models, such as SpeechT5 (Ao et al., 2022) and SpeechNet (Tang et al., 2022), adopt an encoder-decoder framework to handle

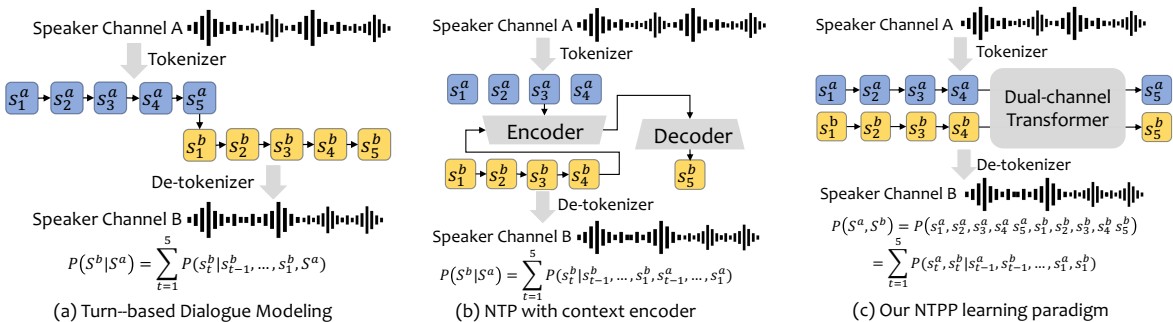

**Figure 2.** An illustration of three different generative models for spoken dialogue is shown: (a) Turn-based dialogue modeling, as formulated in Eq. 4, which is commonly used in cascading and multi-modal approaches; (b) The NTP paradigm with a context encoder architecture, adopted by models such as LSLM, Moshi, and similar variants; (c) Our NTPP paradigm, which employs a decoder-only transformer. Replacing the decoder-only architecture with an encoder-decoder Siamese transformer yields the dGSLM.

various speech-related tasks. However, these approaches require specialized pre-processing and post-processing modules tailored to different input and output modalities. In contrast, models like VioLA (Wang et al., 2023), AudioPaLM (Rubenstein et al., 2023), SpeechGPT (Zhang et al., 2023a), and SpeechGen (Wu et al., 2023) utilize decoder-only transformers, representing both discrete audio and text tokens within a shared vocabulary. Building on these advancements, our work leverages SLMs as pre-trained foundation models, further refining them through continual pre-training on the dual-channel speech.

**Spoken Dialogue Language Models.** Recently, there has been growing interest in spoken dialogue language modeling, inspired by the advancements of GPT-4o (OpenAI, 2024). Works such as LLama-Omni (Fang et al., 2024) and Mini-Omni (Xie & Wu, 2024) generate voice-based dialogue data from text-based question-answering pairs. These SLMs are trained on both speech and text sequences, with inference optimized for parallel processing. However, these approaches resemble multi-modal models (Liu et al., 2023b; Alayrac et al., 2022; OpenAI, 2023) and are not well-suited for handling real-time, streaming spoken interactions. A promising yet relatively underexplored direction is dual-channel speech language modeling. dGSLM (Nguyen et al., 2023) addressed dual-channel speech sequence generation prior to the emergence of modern LLMs, relying on conventional encoder-decoder architectures. In contrast, LSLM (Ma et al., 2024) and Moshi (Défossez et al., 2024) leverage LLMs to model dual-channel speech autoregressively, employing speaker fusion strategies and RQ-transformer (Lee et al., 2022) connections, respectively.

## 3. Preliminaries

**SLMs.** SLMs are usually obtained via continually pre-training LLMs on large-scale single-channel speech sequences. To fit the NTP learning paradigm in LLMs, input

continuous speech signals, $\mathbf{x} \in \mathbb{R}^{T'}$ with time length $T'$, are firstly transformed into a sequence of discrete speech tokens $S = (s_1, ..., s_T) = \mathcal{Q}(\mathbf{x})$ $(T \ll T')$ using a quantizer $\mathcal{Q}$. The quantization operation $\mathcal{Q}$ is mapping each latent feature $\mathbf{f}_i$, where $\mathbf{f} = \mathcal{E}(\mathbf{x}) \in \mathbb{R}^{T \times d}$ is downsampled latent feature of $\mathbf{x}$ derived from an encoder $\mathcal{E}$, to the code index $s_i$ of its nearest embedding vector:

$$ s_i = \arg\min_{v \in [V]} \|\mathbf{z}_v - \mathbf{f}_i\|_2, \tag{1} $$

where $\mathbf{z}_i$ denotes the $i$th embedding vector of the learnable codebook $\mathbf{z} \in \mathbb{R}^{V \times d}$ containing $|V|$ vectors. We refer the readers for the detailed VQ techniques to (van den Oord et al., 2017). If the VQ-tokenizer is adopted, then LLMs are trained via the typical NTP objective:

$$ p(s_1, s_2, ..., s_T) = \prod_{t=1}^{T} p(s_t | s_{t-1}, .., s_2, s_1). \tag{2} $$

Another popular quantization technique used for speech tokenizers is RVQ (Lee et al., 2022) since it can maintain higher reconstruction quality. Specifically, each feature $\mathbf{f}_i$ is estimated by $D$ codes in a residual manner such that $\hat{\mathbf{f}}_i = \mathbf{z}_{i1} + \mathbf{z}_{i2} + ... + \mathbf{z}_{iD}$ where each $\mathbf{z}_{id}$ is indexed $s_{id}$. Therefore, the latent feature is represented by a two-dimensional array of indices $\mathbf{S} \in \mathbb{R}^{T \times D}$. Then the probability distribution $p(\mathbf{S})$ is factorized as

$$ p(s_1, s_2, ..., s_T) = \prod_{t=1}^{T} \prod_{d=1}^{D} p(s_{td} | \mathbf{S}_{<t}, s_{t,<d}). \tag{3} $$

LLMs trained continually using Eq.2 or Eq.3 effectively capture the underlying generative distribution of speech tokens, which are seamlessly integrated into the text vocabulary.

**Generative Spoken Dialogue Modeling.** The spoken dialogue is a pair of speech signals $(\mathbf{x}^a, \mathbf{x}^b)$, containing two speakers' conversations (speakers $a$ and $b$). By applying the

**Training SLM with autoregressive transformers on the dual-channel speech sequence**

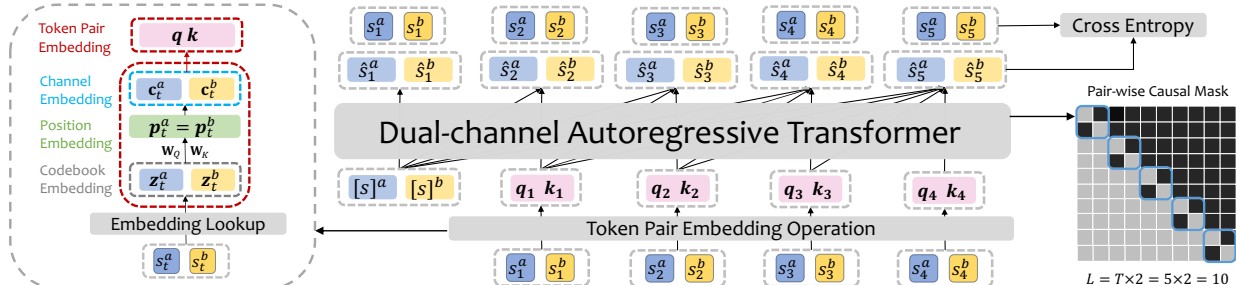

*Figure 3.* The illustration of the autoregressive transformer for learning the dual-channel speech sequence, with the token pair embedding operation (left), the overall architecture (middle) and the pair-wise causal masking mechanism (right).

previously mentioned quantization techniques, $(\mathbf{x}^a, \mathbf{x}^b)$ can be converted to a pair of speech token sequences $(S^a, S^b)$. Existing popular approaches model dialogue in a sequential generation manner, $p(S^b|S^a)$, treating one speaker sequence (assume $a$) as a given condition:

$$p(S^b|S^a) = \prod_{t=1}^{T} p(s_t^b|s_{t-1:1}^b, S^a). \tag{4}$$

As previously mentioned, this approach is limited to handling multi-turn (even one-turn) conversations and cannot support real-time interactions with human users, as it fails to leverage the time-alignment property inherent in human speech conversations. Recently, some works have started to focus on the full duplex capabilities of spoken dialogue language models. Moshi (Défossez et al., 2024) and LSLM (Ma et al., 2024) leverage different architectures following the NTP manner to learn the conditional distribution $p(S^b|S^a)$ of the dual-channel speech:

$$p(S^b|S^a) = \prod_{t=1}^{T} p(s_t^b|s_{t-1:1}^b, s_{t-1:1}^a). \tag{5}$$

Moshi employs the RQ-transformer to encode both $s_{t-1:1}^b$ and $s_{t-1:1}^a$ into a conditional latent representation for predicting $s_t^b$; LSLM, on the other hand, explores various token fusion strategies to merge $s_t^a$ and $s_t^b$ at each time step $t$. Although LSLM adopts a decoder-only architecture, it still models $p(S^b|S^a)$, as its training objective focuses solely on predicting $S^b$. In contrast, dGSLM employs a two-tower Siamese transformer with an encoder-decoder architecture to model the joint distribution $p(S^a, S^b)$, thereby enabling speaker-independent learning. In this work, we achieve the same goal using a decoder-only architecture NTPP. A comparison of these approaches is illustrated in Figure 2.

## 4. Methods: Next-Token-Pair Prediction

### 4.1. NTPP Dual-Channel Generative Modeling

Existing spoken dialogue models are mainly learning the conditional distribution $p(S^b|S^a)$ (or $p(S^a|S^b)$) as shown in Eq. 4 and Eq. 5. In this work, we propose a novel approach, called *next-token-pair prediction (NTPP)*, that explicitly learns the joint distribution of speaker sequences $p(S^a, S^b)$ using the decoder-only transformer. Specifically, the model learns to predict the next token pair $(s_t^a, s_t^b)$ at time step $t$ conditioned on the previously generated token pairs from step $1$ to step $t-1$:

$$
\begin{aligned}
p(S^a, S^b) &= p(s_1^a, s_2^a, ..., s_T^a, s_1^b, s_2^b, ..., s_T^b) \\
&= \prod_{t=1}^{T} p(s_t^a, s_t^b | s_{t-1}^a, ..., s_2^a, s_1^a, s_{t-1}^b, ..., s_2^b, s_1^b).
\end{aligned}
\tag{6}
$$

Unlike Eq. 4 and Eq. 5, the above Eq. 6 is learning to predict both $s_t^a$ and $s_t^b$ during the training process. Then we decompose the distribution $p(s_t^a, s_t^b | s_{t-1}^a, s_{t-1}^b, ..., s_1^a, s_1^b)$ by assuming a conditional independence between $s_t^a$ and $s_t^b$ at each step $t$ such that

$$
\begin{aligned}
p(s_t^a, s_t^b | s_{t-1}^a, ..., s_2^a, s_1^a, s_{t-1}^b, ..., s_2^b, s_1^b) &= \\
p(s_t^a | s_{t-1:1}^a, s_{t-1:1}^b) p(s_t^b | s_{t-1:1}^a, s_{t-1:1}^b).
\end{aligned}
\tag{7}
$$

We illustrate this conditional independence and the dialogue distribution modeling in Figure 2. The probability distribution in Eq.6 and Eq.7 adheres to a fundamental inductive bias that *a person's speech is influenced by both his own previous statements and what he has heard in the past.* This approach naturally incorporates mutual dependence, as both $p(s_t^a | s_{t-1:1}^a, s_{t-1:1}^b)$ and $p(s_t^b | s_{t-1:1}^a, s_{t-1:1}^b)$ are modeled. Additionally, the time-alignment property of the two speaker sequences is preserved through the joint prediction of $(s_t^a, s_t^b)$ at each time step $t$.

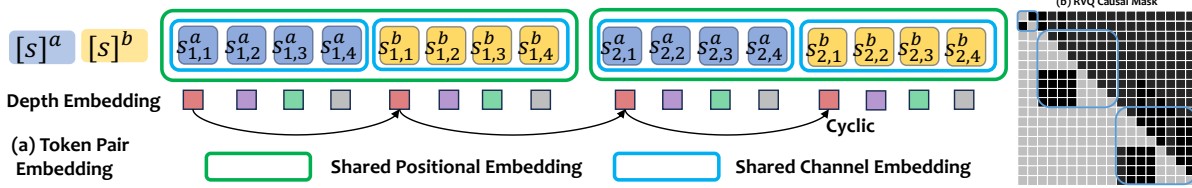

*Figure 4.* The illustration of two modified components of RVQ-based dual-channel transformer: (a) Token pair embedding operation (including cyclic depth embedding) and (b) RVQ causal masking mechanism.

### 4.2. Autoregressive Dual-channel Speech Transformer

The remaining challenge is how the model can learn $p(S^a, S^b)$ as defined in Eq. 6 and Eq. 7 in *decoder-only architectures*. To address this, we propose the autoregressive dual-channel speech transformer. The two speech sequences are rearranged in an interleaved order: $S = ((s_t^a, s_t^b), (s_{t-1}^a, s_{t-1}^b), ..., (s_1^a, s_1^b))$. At each time step $t$, the model predicts a pair of tokens $S_t = (s_t^a, s_t^b)$. This design requires only minimal modifications to adapt the decoder-only transformer architecture of LLMs. Specifically, two essential components are adjusted to accommodate the sequence of token pairs: the token pair embedding operation and the pair-wise causal attention masking mechanism.

**Token Pair Embedding.** The token pair embedding operation is used to transform each index token $s_t$ back to continuous latent embedding. For token pair $S_t$, there are three important latent embeddings: codebook embedding $\mathbf{z}_t$, positional embedding $\mathbf{p}_t$ and channel embedding $\mathbf{c}_t$. $\mathbf{z}_t^a$ and $\mathbf{z}_t^b$ can be easily retrieved from the codebook $Z$ by querying token indices $s_t^a, s_t^b$ respectively. For positional embedding, we inherit the rotary positional encoding (Su et al., 2024) to indicate which time step that each token belongs to. Note that each token pair $s_t^a$ and $s_t^b$ share the same positional embedding such that $\mathbf{p}_t^a = \mathbf{p}_t^b$ since they are aligned at the same time step $t$. Compared to the conventional SLM architecture, we have two aligned speech sequences instead of one single sequence. Hence, to inform the model about the speaker role of each sequence, we additionally include a channel embedding $\mathbf{c}_t$ for each token to help the model distinguish which speaker generates the token. $\mathbf{c}_t$ is a simple one-hot encoding $\mathbf{c}_t = \textit{one-hot}(\mathbf{id})$, where $\mathbf{id}$ is either $a$ or $b$. Following the implementation of Llama 3 (Dubey et al., 2024), we add both positional embedding and channel embedding to the query $\mathbf{q}$ and key vectors $\mathbf{k}$ derived in the attention mechanism. Then the token embedding operation for each token pair $(s_t^a, s_t^b)$ is as follows:

$$\mathbf{q} = \mathbf{W}_Q[\mathbf{z}_t^a, \mathbf{z}_t^b] + [\mathbf{p}_t^a, \mathbf{p}_t^b] + [\mathbf{c}_t^a, \mathbf{c}_t^b], \qquad (8)$$

$$\mathbf{k} = \mathbf{W}_K[\mathbf{z}_t^a, \mathbf{z}_t^b] + [\mathbf{p}_t^a, \mathbf{p}_t^b] + [\mathbf{c}_t^a, \mathbf{c}_t^b], \qquad (9)$$

where $\mathbf{W}_Q$ and $\mathbf{W}_K$ are projection matrices for queries $\mathbf{q}$ and keys $\mathbf{k}$ respectively.

**Pair-wise Causal Masking.** In the standard LLM attention mechanism, causal masking ensures that each token can only attend to previous tokens, preventing any access to future tokens. Consequently, the masking matrix $\mathbf{M}$ is structured as a lower-triangular matrix. In our dual-channel sequence setting, the key distinction lies in the diagonal of the masking matrix. Specifically, the $2 \times 2$ block-wise diagonal entries $\mathbf{m}$ in $\mathbf{M} \in \mathbb{R}^{2T \times 2T}$ follow a pair-wise causal masking strategy. Within each block $\mathbf{m}$, only the diagonal entries remain unmasked, while all other entries are masked. This enforces the constraint that $s_t^a$ and $s_t^b$ cannot attend to each other at any given time step $t$.

The solution described above applies to the simple VQ tokenizer case, as illustrated in Figure 3. We omit the training loss function here, as it is nearly identical to the NTP training loss, with the only difference being the inclusion of an additional loss term for predicting the second speech channel sequence.

### 4.3. Generalizing Solutions to RVQ Tokenizers

As mentioned in the preliminary section, the RVQ-tokenizer is widely adopted in SLMs for achieving higher reconstruction quality. It is a non-trivial challenge to extend our solutions to the RVQ-tokenizer since each speech sequence becomes a 2D array of codes $\mathbf{S} \in \mathbb{R}^{T \times D}$ instead of a one-dimensional index token sequence. How to maintain the decoder-only transformer architecture to learn $p(\mathbf{S}^a, \mathbf{S}^b)$ remains a tricky issue. To solve this issue, we flatten $\mathbf{S}$ into a one-dimensional sequence $S^{a/b} = ((s_{1,1}^{a/b}, ..., s_{1,D}^{a/b}), (s_{2,1}^{a/b}, ..., s_{2,D}^{a/b}), ..., (s_{T,1}^{a/b}, ..., s_{T,D}^{a/b}))$ (either $a$ or $b$) with sequence length $T \times D$. In this way, the dual-channel speech sequence can be re-arranged as

$$S = (\mathbf{S}_1^a, \mathbf{S}_1^b, ..., \mathbf{S}_T^a, \mathbf{S}_T^b) = ((s_{1,1}^a, ..., s_{1,D}^a),$$
$$(s_{1,1}^b, ..., s_{1,D}^b), ..., (s_{T,1}^a, ..., s_{T,D}^a), (s_{T,1}^b, ..., s_{T,D}^b)).$$
$$(10)$$

This sequence is similar to the VQ-based interleaving sequence and just additionally contains $D$ residual depth tokens for each time step $t$ and each speaker channel.

**RVQ Token Pair Embedding.** The codebook embedding $\mathbf{z}_t$, the channel embedding $\mathbf{c}_t$ and the positional embedding

$\mathbf{p}_t$ remain the same operation as VQ-based solution. The same $\mathbf{p}_t$ is shared by $\mathbf{S}_t^a$ and $\mathbf{S}_t^b$. The same channel embedding $\mathbf{c}_t$ is shared by $s_{t,d}$ for all $d$. One challenging problem brought by RVQ-tokenizer is how to identify the depth of each token. For example, when the $i$th token is input to the model, how could the model know the depth of the token (range from 1 to $D$)? To alleviate this issue, we introduce the *cyclic depth embedding* $\mathbf{d}$ as follows:

$$\mathbf{d} = (\sin((2\pi * i)/D), \cos((2\pi * i)/D)) \qquad (11)$$

Note that this embedding is cycled with step length $D$. That is, $\mathbf{d}_i = \mathbf{d}_{i+D}$ for every position $i$ (In VQ-tokenized sequences, $i = t$; While in RVQ-tokenized sequences, $t = i/(2D)$ due to $D$ depth tokens for each channel).

**RVQ Causal Masking.** The causal masking strategy for RVQ-based tokenized sequences closely resembles that of VQ-based tokenized sequences. However, since the depth increases from 1 to $D$ (transitioning from VQ to RVQ), the $2D \times 2D$ block-wise diagonal entries $\mathbf{m}$ of the masking matrix $\mathbf{M} \in \mathbb{R}^{(TD)\times(TD)}$ are specifically adjusted. The upper triangular part of $\mathbf{m}$ remains masked to ensure that current tokens cannot attend to future tokens—this includes preventing shallow-depth tokens $s_{t,d}$ from attending to deeper-depth tokens $s_{t,>d}$ for each step $t$ and each channel). Additionally, the bottom-left $D \times D$ submatrix is masked to ensure that $s_t^a$ and $s_t^b$ do not attend to each other.

The extended solutions in the section 4.3 RVQ-tokenized sequences are illustrated in Figure 4. Note that we omit the special start tokens during discussions for simplicity.

# 5. Experiments

## 5.1. Dataset and Baselines

**Dataset.** Our NTPP is trained using a *two-stage pipeline*. In the first stage, we establish the SLM with foundational speech capabilities by training the model on three speech datasets, totaling approximately 140,000 hours. This phase focuses on both speech understanding and synthesis. Unlike other models (Fang et al., 2024; Xie & Wu, 2024) that require additional text alignments, our approach follows a textless learning paradigm. This eliminates the need for speech-text alignment, reducing data preprocessing requirements and significantly increasing the amount of available training data. In the second stage, we equip our SLMs with the ability to listen and speak simultaneously through NTPP training. For this, we leverage the Fisher dataset (Cieri et al., 2004), which contains 2,200 hours of phone conversations between randomly paired participants discussing predefined topics. A key advantage of the Fisher dataset is that each speaker's audio is recorded on separate channels, providing high-quality dual-channel speech streams essential for NTPP training. Since the original audio is sampled at 8kHz,

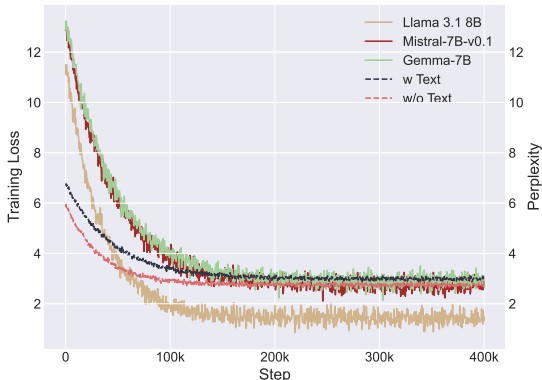

*Figure 5.* Comparison of training loss curves across different models. Solid lines show the training progress for different foundation models. Dashed lines represent an ablation study comparing a model trained with textual data (w Text) versus without (w/o Text).

we use Librosa [1] to upsample it to 16kHz for consistency with our training setup.

**Baselines** We evaluate NTPP's performance by comparing it with established generative speech systems. In terms of turn-taking statistics within generated dialogues, we compared NTPP specifically with dGSLM (Nguyen et al., 2023), as it represents a comparable full-duplex generative model trained on the Fisher dataset. Following dGSLM's setting (Nguyen et al., 2023), we include a cascaded baseline which consists of an Automatic Speech Recognition(ASR) model, followed by a text-based language model and a Text-To-Speech (TTS) module. Following the settings of (Nguyen et al., 2023), we select wav2vec2-large(Baevski et al., 2020), KenLM(Heafield, 2011), and Google TTS API as the modules respectively. For assessing the meaningfulness and naturalness of the interaction, we extend our comparison to include Moshi (Défossez et al., 2024) and SyncLLM (Pope et al., 2023a). Notably, SyncLLM is a full-duplex dialogue agent designed for real-time, overlapping speech interactions through the joint, streaming processing of speech input and output.

## 5.2. SLM Training & Implementation Details

**Audio Tokenizer & Token Vocoder.** We train an RVQ speech tokenizer based on (Zeghidour et al., 2022), which encodes each second of audio into 40 discrete tokens from a codebook of size 4096. Due to the limitations of the single-speaker token vocoder presented in (Kong et al., 2020), we train a multi-speaker HiFi-GAN to decode speech signals from discrete tokens. The HiFi-GAN architecture consists of a generator and multiple discriminators. The generator uses look-up tables to embed discrete representations and the embedding sequences are up-sampled by a series of

---

[1] https://librosa.org/doc.

*Table 2.* Turn-taking statistics (event occurrences and durations/min) for generated dialogues, compared to ground truth using mean absolute differences ($|\Delta|$) between predicted and ground truth values. NTPP model results cover different temperature settings [0.1, 0.5, 0.9]. Lower $|\Delta|$ values indicate better performance, with the best results highlighted in bold.

| Model | Number of occurrences / min | | | | Cumulated duration /min | | | |
|---|---|---|---|---|---|---|---|---|
| | $|\Delta\textbf{IPU}|$ | $|\Delta\textbf{Pause}|$ | $|\Delta\textbf{Gap}|$ | $|\Delta\textbf{Overlap}|$ | $|\Delta\textbf{IPU}|$ | $|\Delta\textbf{Pause}|$ | $|\Delta\textbf{Gap}|$ | $|\Delta\textbf{Overlap}|$ |
| dGSLM w/o CA | 3.9 | 2.9 | 3.6 | 1. | 12.1 | 8.3 | 1.4 | 2.5 |
| dGSLM | 1.6 | 3.4 | 2. | 2.9 | 4.6 | 3.6 | 1.8 | 1.9 |
| LSLM | 2.2 | 3.6 | 2.4 | 3.2 | 4.1 | 3.4 | 1.5 | 2.3 |
| Cascaded | 4.1 | 7.0 | 7.4 | 6.5 | 4.3 | 5.5 | 0.9 | 3.6 |
| $NTPP_{0.1}$ | 1.4 | 2.1 | 2.0 | 1. | 3.2 | **2.5** | 1.2 | 2.1 |
| $NTPP_{0.5}$ | 1.5 | **1.9** | 1.8 | 1.5 | **2.9** | 3.0 | **0.9** | 2.2 |
| $NTPP_{0.9}$ | **1.3** | 2.3 | **1.5** | **0.9** | 3.3 | 2.8 | 1.4 | **1.9** |

blocks composed of transposed convolution and a residual block with dilated layers. The speaker embedding is concatenated to each frame in the up-sampled sequence. The discriminator features a Multi-Period Discriminator and a Multi-Scale Discriminator, which have the same architecture as (Kong et al., 2020).

**LLM Backbones.** We leverage three well-known LLMs as the backbone of our SLM: LLaMA 3.1–8B (Dubey et al., 2024), Mistral-7B-v0.1 (Jiang et al., 2023a), and Gemma-2-9B (Team et al., 2024). We evaluate the performance of SLMs obtained by training these LLMs with the NTP objective on a large-scale single-channel audio dataset. Perplexity on the test set is used as the evaluation metric. Figure 5 shows the training loss curves over time for all three models. Each model demonstrates a consistent downward trend, indicating effective learning. Mistral-7B and Gemma show comparable learning dynamics, while LLaMA 3.1—known for its strong reasoning capabilities in text—achieves lower training loss more quickly. This finding supports our hypothesis that stronger text-based models serve as more effective initializations for continual speech learning, consistent with the perspective of treating audio as a new language.

Perplexity curves highlights that during the pre-training phase, the NTPP model demonstrates superior performance when trained exclusively on audio data ("w/o Text") compared to when trained with both audio and its ASR text transcriptions ("w Text"). Specifically, the audio-only approach leads to significantly faster convergence and consistently lower perplexity—a measure indicating better predictive capability of the model. The results show that the audio-only setting leads to significantly faster convergence and consistently lower perplexity throughout training. This indicates that eliminating the ASR transcript—which may introduce recognition errors or redundancy—allows the model to focus on more informative acoustic cues, thereby facilitating more stable and efficient learning. In contrast, the inclusion of textual supervision appears to slow optimization and result in higher perplexity, suggesting potential modality interference.

### 5.3. Turn-Taking Event Distribution

Following the experimental setting in dGSLM (Nguyen et al., 2023), we evaluate the dialogue systems with turn-taking capabilities using corpus-level statistics (Ardila et al., 2019) and testing on the Fisher dataset (Cieri et al., 2004). We evaluate the linguistic quality and turn-taking dynamics of generated dialogues using various models, as shown in Table 2. The detailed evaluation settings are in the Appendix C.2. LSLM (Ma et al., 2024) integrates speaker channels at the embedding layer and only predicts the speech tokens from the assistant channel, demonstrating a notable reduction in the number of Inter-Pausal Units (IPUs) and gaps, indicating smoother transitions between speakers. The dGSLM (Nguyen et al., 2023), particularly with the cross-attention module, shows a significant decrease in the cumulative duration of pauses and gaps, suggesting more fluid and continuous dialogue. Comparatively, NTPP exhibits balanced performance with moderate reductions in both the number and duration of turn-taking events, highlighting their potential for generating natural and coherent dialogues. These findings underscore the importance of model architecture in optimizing dialogue flow.

### 5.4. Interruptions & Reflective Pause

We further develop a comprehensive evaluation framework comprising 400 diverse conversational scenarios specifically designed to systematically capture natural dialogue dynamics, with an emphasis on pauses and interruptions. These scenarios are carefully crafted using GPT-4 to reflect the complexity and nuance of human interactions. We then use ChatTTS(cha, 2024) to synthesize high-quality speech from the generated text, effectively mimicking the acoustic characteristics of real-world conversations. As shown in Figure 6, NTPP demonstrates closer alignment with human reference judgments, which are meticulously established through manual annotation by human evaluators, in both

speaking-up and interruption scenarios when acting as a listener. For instance, informing its response strategy, NTPP employs Voice Activity Detection (VAD) and considers a silence state to have been reached after 200ms of continuous non-speech is detected, facilitating its decisions on when to speak or acknowledge a pause. This suggests that NTPP more effectively captures the subtleties of human conversational behavior compared to other models.

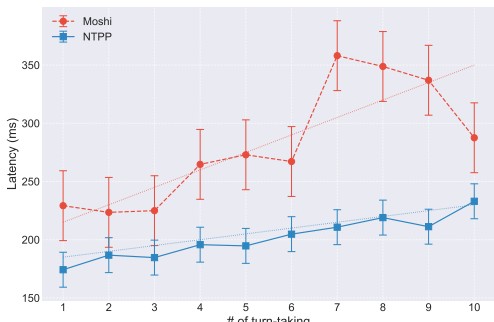

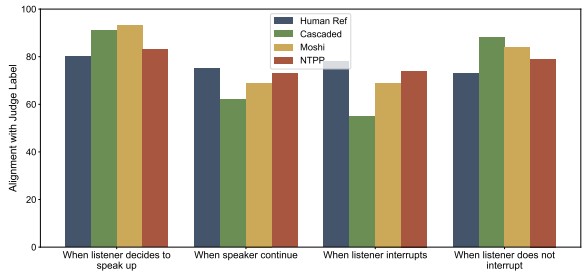

*Figure 6.* NTPP, when acting as a listener, shows closer alignment with Human Reference judgments in both speaking up and interrupting scenarios.

### 5.5. Human Evaluation

We follow the evaluation protocol from (Veluri et al., 2024) and conduct a human study involving 25 annotators with native-level English proficiency. Adopting the Mean Opinion Score (MOS) framework, we use a 5-point Likert scale to evaluate the Naturalness (N-MOS) of turn-taking and the Meaningfulness (M-MOS) of the generated dialogue content. Table 3 presents a comparison of NTPP with various baselines in terms of both naturalness and meaningfulness. We also include performance comparisons on the out-of-distribution Candor test set (Reece et al., 2023) to assess generalization.

### 5.6. Inference Latency

In multi-turn interactions, latency is a crucial metric for assessing the performance of speech interaction models, as it captures the time between receiving the end of input speech and the start of output speech, directly impacting user experience. As shown in Figure 7, our NTPP consistently achieves lower inference latency than Moshi, especially as the number of turn-taking rounds increases. We attribute this advantage to NTPP's efficient memory usage: it maintains a single KV Cache, while Moshi requires two separate KV Caches for the two separate transformers. This difference becomes increasingly significant as conversational context grows longer.

*Figure 7.* Our method (blue) demonstrates lower latency growth compared to Moshi's linear degradation(red), maintaining response times below perceptual thresholds (220 ms) across all rounds.

### 5.7. Speaker Independence

To assess the speaker-independence of various models, we conduct a speaker-swapped evaluation. All models are first trained on the dual-channel Fisher dataset (Cieri et al., 2004) using the canonical speaker order. We reverse the input speaker sequence without any additional fine-tuning or adaptation. This setup allows us to test whether a model's performance remains stable when the roles of the two speakers are exchanged—an essential indicator of robustness and generalization in dialogue turn-taking modeling. We evaluate this on *both the training and test sets*. As shown in Table 4, both dGSLM and NTPP exhibit minimal variation in key turn-taking metrics under speaker-swapped conditions—nearly zero variation on the training set and consistently low variation on the test set—demonstrating strong speaker-independent behavior. Moshi shows substantial deviations in metrics such as the number and duration of IPUs, pauses, gaps, and overlaps. These findings suggest that Moshi relies on speaker-conditioned generation (due to its modeling of conditional distributions), resulting in degraded performance when the input speaker order is reversed.

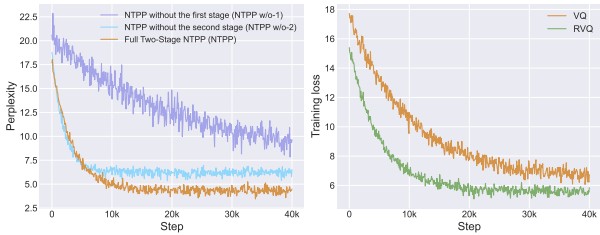

*Figure 8.* Comparative performance analysis. (a) Ablation study of NTPP model training stages, illustrating perplexity versus training steps for the full two-stage model and variants lacking either stage one or stage two. For better visualization, we trim the stage one training steps to be the same as stage two. (b) Comparison of VQ and RVQ model training loss as a function of training steps.

*Table 3.* Human evaluation results across different SLMs. Meaningfulness (Meaning.) and Naturalness (Nat.) scores (ranging from 1 to 5) represent mean estimates and their standard errors (shown in parentheses), reported both overall and separately for the Fisher and CANDOR datasets.

| Model | Overall | | Fisher | | CANDOR | |
|---|---|---|---|---|---|---|
| | Meaning. ↑ | Nat. ↑ | Meaning. ↑ | Nat. ↑ | Meaning. ↑ | Nat. ↑ |
| dGSLM | 1.38 (0.10) | 3.85 (0.12) | 1.82 (0.09) | 4.10 (0.13) | 1.51 (0.12) | 2.85 (0.18) |
| SyncLLM | 3.85 (0.06) | 4.10 (0.05) | 4.10 (0.04) | 4.33 (0.08) | 3.85 (0.09) | 3.91 (0.08) |
| Moshi | 3.90 (0.07) | 3.95 (0.06) | 3.20 (0.10) | 3.90 (0.08) | 3.95 (0.08) | 3.95 (0.08) |
| **NTPP** | **3.95 (0.04)** | **4.15 (0.06)** | **4.10 (0.06)** | **4.42 (0.06)** | **4.05 (0.04)** | **4.05 (0.10)** |
| GT | 4.90 (0.01) | 4.95 (0.02) | 4.90 (0.03) | 4.90 (0.04) | 4.90 (0.02) | 4.95 (0.02) |

*Table 4.* Linguistic quality and turn-taking metrics under speaker-swapped inference. Reported values denote the absolute difference between deviation metrics $\Delta M$ (e.g., $\Delta$IPU) under the original and swapped speaker orders, computed as $|\Delta M_{\text{original}} - \Delta M_{\text{swapped}}|$. Lower values indicate higher robustness to speaker order permutation.

| Split | Model | Number of Occurrences / min | | | | Cumulated Duration / min | | | |
|---|---|---|---|---|---|---|---|---|---|
| | | \|$\Delta$**IPU**\| | \|$\Delta$**Pause**\| | \|$\Delta$**Gap**\| | \|$\Delta$**Overlap**\| | \|$\Delta$**IPU**\| | \|$\Delta$**Pause**\| | \|$\Delta$**Gap**\| | \|$\Delta$**Overlap**\| |
| **Train** | dGSLM | 0.05 | 0.14 | **0.04** | 0.06 | 0.09 | **0.15** | **0.11** | 0.09 |
| | Moshi | 0.43 | 0.32 | 0.29 | 0.29 | 0.30 | 0.56 | 0.55 | 0.58 |
| | *NTPP* | **0.03** | **0.07** | 0.07 | **0.05** | **0.06** | 0.18 | 0.14 | **0.06** |
| **Test** | dGSLM | 0.20 | 0.15 | **0.18** | 0.21 | 0.39 | **0.32** | **0.41** | 0.32 |
| | Moshi | 0.43 | 0.38 | 0.37 | 0.39 | 0.57 | 0.62 | 0.84 | 0.68 |
| | *NTPP* | **0.18** | **0.14** | 0.19 | **0.20** | **0.35** | 0.38 | 0.45 | **0.24** |

### 5.8. Ablation Studies

We further investigate the importance of two-stage training and the use of an RVQ tokenizer in this section. As shown in Figure 8 (a), our two-stage training strategy consistently achieves lower perplexity compared to the one-stage approach, which omits pretraining on single-channel audio. This suggests that pretraining on single-channel audio provides a strong foundation, significantly improving performance on subsequent dual-channel speech learning. As expected, omitting the second-stage NTPP training on dual-channel speech also leads to performance degradation. Figure 8 (b) compares training loss between VQ and RVQ tokenizers, with RVQ yielding consistently lower loss, highlighting the importance of developing extended solutions tailored to this tokenizer.

### 6. Limitations and Future Works

One major limitation is the limited availability of large-scale dual-channel speech data. Unlike single-channel audio, which can be sourced from the vast amount of open-source data available online, dual-channel speech data requires either additional channel separation operations on single-channel audio or meticulous collection from real-world human conversations. We hope our work will inspire the community to gather large-scale dual-channel or even multi-channel spoken dialogue datasets. In the future, we

plan to explore synthetic data strategies for generating high-quality dual-channel speech data.

### 7. Conclusion

In this work, we introduce a novel spoken dialogue generative model based on the NTPP paradigm. To effectively capture the dynamics of human conversations, we design a decoder-only dual-channel transformer that models the joint distribution of two speaker channels. Our approach includes both VQ-tokenizer and RVQ-tokenizer versions, significantly enhancing the real-time conversational capabilities of diverse SLMs. Through extensive evaluations across multiple benchmarks, we demonstrate the effectiveness and superiority of our method in generating natural and coherent spoken dialogues, paving the way for more advanced and interactive speech-based AI systems.

### Impact Statement

This paper aims to advance the field of SLMs. By enhancing SLMs through our approach, we enable more natural and seamless spoken interactions with human users, benefiting various domains such as voice-based personal assistants, customer service chatbots, and online education with voice interactions. However, a potential negative societal consequence is the risk of misuse in telecom fraud, as our

approach improves the naturalness of AI-generated conversations. To mitigate this risk, further advancements in AI safety techniques and fraud detection systems are necessary.

## Acknowledgements

The research is supported by the Tencent Al Lab RBFR2024004.

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

## A. Other Related Works

**Autoregressive Generative Models.** The autoregressive generative modeling has achieved remarkable success in natural language processing, giving rise to a variety of powerful LLMs (Sutskever et al., 2014; OpenAI, 2024; 2023; Patel et al., 2023). Inspired by these LLMs, numerous studies have examined the application of autoregressive modeling in other domains, such as images (van den Oord et al., 2017; Esser et al., 2021; Li et al., 2024; Tian et al., 2024; Lee et al., 2022; Chang et al., 2022), graphs (You et al., 2018), videos (Weissenborn et al., 2020), molecules (Shi et al., 2020; Schwaller et al., 2019) and protein sequences (Madani et al., 2023; Lin et al., 2022). The core idea of autoregressive modeling is to iteratively generate an entire sequence by predicting each part from the preceding segments, a property that makes it particularly well-suited for audio generation and inherently strong in reasoning tasks (Liu et al., 2023c; 2024).

**Multi-Modal LLMs.** Multi-modal Large Language Models (MM-LLMs) strive to incorporate knowledge from diverse modalities. A key category of MM-LLMs concentrates on developing connectors (Li et al., 2022; 2023; Liu et al., 2023b; Alayrac et al., 2022) that identify knowledge alignment across various modalities. An alternative strategy (Team, 2024; Zhou et al., 2024; Xie et al., 2024) merges all modalities into a cohesive sequence of tokens and utilizes LLMs to sequentially generate them using modified attention masks. These methods (Wu et al., 2024; Su et al., 2023; Fu et al., 2024) even integrate audio as an input modality, and by simply combining text and audio through MM-LLM techniques, they can address one-direction conditional generation tasks such as speech-to-text translation (e.g., ASR and spoken language understanding) (Radford et al., 2023; Zhang et al., 2023b; Deshmukh et al., 2023; Arora et al., 2023; Tang et al., 2024; Chu et al., 2024; Zhou et al., 2023; Baevski et al., 2020; Gao et al., 2023) and text-to-speech translation (e.g., TTS) (Elizalde et al., 2023; Liu et al., 2023a; Huang et al., 2023; Nachmani et al., 2023; Yang et al., 2023; Kreuk et al., 2023; Borsos et al., 2023; Copet et al., 2023; Chen et al., 2024; Anastassiou et al., 2024; Jiang et al., 2023b; Kong et al., 2021; Shen et al., 2024; Casanova et al., 2022; Siuzdak, 2024; Yang et al., 2024; Kharitonov et al., 2023; Le et al., 2023). However, these methods are limited to multi-turn (or even single-turn) QA tasks (where the model produces an answer only after the question is completed, as signaled by the voice-activity-detection (VAD) module, e.g. special tokens, button pressing and hard tunr-taking interval threshold.) and thereby struggle with real-time voice interaction tasks, which is the primary focus of our work.

## B. Streaming Conditional Inference

In order to simulate a real-time user-assistant communication scenario, our speech LMs improved by NTPP should be proficient in conducting conditional inference with streaming user voice input. In this inference setting, one speaker's voice input is provided as the user, and the model is assigned the task of inferring the other audio channel. This creates a situation that resembles a constrained generation problem. If the inference process strictly follows the training process, then the model should predict $\hat{s}_t^b$ immediately after receiving the speaker's voice input $s_t^a$ at time $t$. However, due to the VQ-VAE tokenization mechanism, it's not feasible to receive just a single speech token from the speaker channel during the streaming inference. This is because the VQ-VAE requires a complete continuous speech signal input with a specific time length. Therefore, we need to determine when the model should start generating spoken responses upon receiving streaming user input speech tokens. Specifically, the inference process follows the chunk-wise style, con-

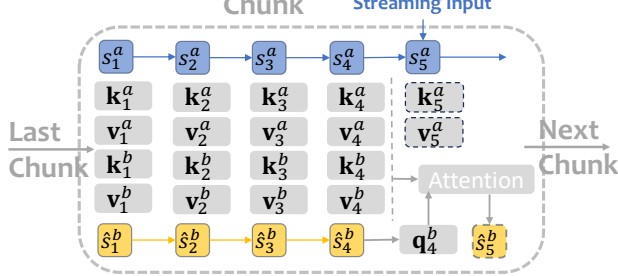

*Figure 9.* The figure illustrates the chunk-wise streaming inference process. Within each chunk, $(s_1^a, s_2^a, s_3^a, s_4^a, s_5^a)$ represents the provided speaker sequence. Their corresponding keys and values are stored in the KV-cache. NTPP sequentially predicts tokens $(\hat{s}_1^b, \hat{s}_2^b, \hat{s}_3^b, \hat{s}_4^b, \hat{s}_5^b)$ based on generated query vectors, which are directed to the Key-Value (KV) cache through attention computations. Once a chunk is filled, the inference proceeds to the next chunk.

taining a predetermined $\lambda$ number of tokens. As long as the number of user input tokens reaches $\lambda$ (a chunk of speaker input is given), our model begins to generate predictions until the number of predicted tokens also reaches $\lambda$ (a chunk is filled). This procedure is repeated until the end of user voice inputs (e.g., the conclusion of the voice-assistant service). Here we omit the RVQ-verision inference mechanism since the only difference is the inclusion of additional depth tokens (if $\lambda = 5$ in the VQ-based inference, then $\lambda = 5D$ in the RVQ-based inference).

# C. More Implementation Details and Hyper-Parameter Settings

## C.1. Hyper-Parameter Settings

Our model is trained on 16 A100 GPUs, utilizing a cosine annealing learning rate scheduler with a minimum learning rate of 4e-6 and a maximum learning rate of 4e-4. Each training epoch consists of 40,000 steps, with batch size 64 for each step. During fine-tuning, we use learn rate from 4e-6 to 5e-5.

## C.2. Dialogue Turn-Taking Statistics Evaluation

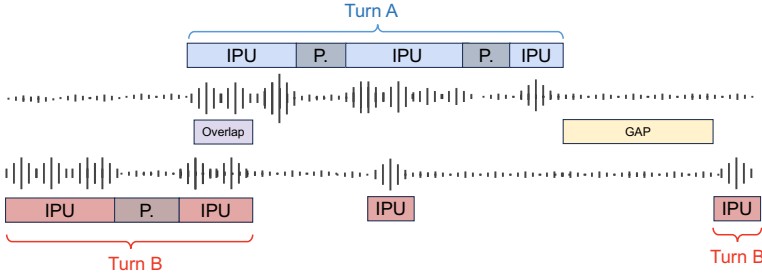

*Figure 10.* Illustration of turn-taking events: IPU (Inter-Pausal Unit), Turn (for Speaker A and Speaker B, resp), P.(Within-Speaker Pause), Gap and Overlap.

Our model generates two audio channels at the same time, allowing us to use basic Voice Activity Detection (VAD) tools on the output to gather turn-taking metrics. According to the settings in (Nguyen et al., 2023), an Inter-Pausal Unit (IPU) is a continuous speech segment within one speaker's channel, bordered by VAD-detected silences longer than 200ms on both ends. Silence is defined as the lack of voice signals on either channel, while overlap refers to segments where voice signals are detected on both channels. Silences can be further divided into gaps (between IPUs of different speakers) and pauses (within the same speaker's IPUs). Consecutive IPUs by the same speaker, separated by a pause, are merged into a single turn. Our analysis will focus on measuring the duration distribution of IPUs, gaps, pauses, and overlaps in both the training corpus and the dialogues generated by our various models.

## C.3. Reflective Pause Audio Dataset

**Prompt for reflective pause**

"Hmm..., this question is a bit complicated, I need to think about it."
"Let me recall, uh..., yes, we went to the park that day."
"You know, that..., oh, yes, it's the new restaurant."
"I remember he mentioned it, um..., it seems to be last Friday."
"This matter, um..., I think we need to discuss it again."
"Let me think about it, uh..., yes, that's it."
"I'm not sure, um..., maybe I need to confirm it again."
"This question, um..., I think we can solve it this way."
"Let me think about it again, uh..., yes, I remember it."
"The one you mentioned, um..., I seem to have some impression."
"We need to deal with the budget issue of this project. Um..., this problem is a bit complicated, I need to think about it."
"Do you remember the last time we met? Let me recall, uh..., yes, we went to the park that day."
"Have you heard about the new restaurant? You know, that..., oh, yes, that new restaurant."
"When did he tell you the news? I remember he mentioned it, uh..., it seems to be last Friday."
"Do you have any suggestions about this plan? This matter, uh..., I think we need to discuss it again."
"Can you give me an example? Let me think about it, uh..., yes, that's it."
"Are you sure this data is correct? I'm not sure, uh..., I may need to confirm it again."
"How should we deal with this emergency? This problem, uh..., I think we can solve it this way."
"Can you explain this concept again? Let me think about it again, uh..., yes, I remember it."
"Do you know what he is talking about? The one you said, uh..., I seem to have some impression."

---

**Prompt for GPT score**

Content (1-5 points):
1 point: The response is largely irrelevant, incorrect, or fails to address the user's query. It may be off-topic or provide incorrect information.
2 points: The response is somewhat relevant but lacks accuracy or completeness. It may only partially answer the user's question or include extraneous information.
3 points: The response is relevant and mostly accurate, but it may lack conciseness or include unnecessary details that don't contribute to the main point.
4 points: The response is relevant, accurate, and concise, providing a clear answer to the user's question without unnecessary elaboration.
5 points: The response is exceptionally relevant, accurate, and to the point. It directly addresses the user's query in a highly effective and efficient manner, providing exactly the information needed.

Style (1-5 points):
1 point: The response is poorly suited for speech interaction, possibly including structured elements like lists or being overly complex, disjointed, or difficult to understand.
2 points: The response is somewhat suitable but may be too long, too short, or awkwardly phrased, making it less effective in a speech interaction context.
3 points: The response is generally suitable for speech interaction, but it may have minor issues with length, clarity, or fluency that detract slightly from the overall effectiveness.
4 points: The response is well-suited for speech interaction, with appropriate length, clear language, and a natural flow. It is easy to understand when spoken aloud.
5 points: The response is perfectly suited for speech interaction. It is the ideal length, highly clear, and flows naturally, making it easy to follow and understand when spoken.

Below are the transcription of user's instruction and models' response:
### [Instruction]: **{instruction}**
### [Response]: **{response}**

After evaluating, please output the scores in JSON format: {"content": content score, "style": style score}. You don't need to provide any explanations.

---

# D. Case Study

```
Scenario: A user engages in a conversation with Parrot, describing an
    object and asking the model to identify it.
User: Please listen to my description of an object below, and say its
    name when you have guessed it. The description is: it has four legs,
    a flat surface, and is often used for dining or working...
Parrot: I guess it might be a table.
```

*Figure 11.* Case study of NTPP interrupt human speaking correctly and timely.

To intuitively understand the differences in responses from our models, we provide an example in Figure 11. In this scenario, NTPP interrupts the user at the precise moment it has gathered enough information to make an accurate prediction. This capability is a significant departure from current models that would typically wait for the user to finish speaking before responding. The ability to interject appropriately not only demonstrates the model's advanced comprehension skills but also enhances the fluidity and naturalness of the interaction.

