# OpenReview forum: "NTPP: Generative Speech Language Modeling for Dual-Channel Spoken Dialogue via Next-Token-Pair Prediction"
_ICML.cc/2025/Conference — ICML 2025 poster_

### Official Review · Reviewer_Ubh3 · 2025-03-07

**Overall Recommendation:** 3

**Summary:**

This paper introduces PARROT, a system designed to handle dual-channel spoken dialog using large language models (LLMs). Authors highlight importance of capturing conversational features such as overlaps, pauses, and interruptions, to provide more realistic spoken interactions.
Building upon previous work like dGSLM and Moshi, the authors explore how to utilize dual-channel speech data within modern LLMs.
The core of their approach is a Next-Token-Pair Prediction (NTPP) paradigm, where a decoder-only transformer is trained to predict the next pair of speech tokens based on past dialog (it predicts at time t the next token pair (a,b) of both channels: user channel and bot channel). Their dual-channel transformer architecture can also benefit from recent KV-cache optimizations for lower inference latency.
They compare PARROT with existing methods (dGSLM & Moshi) along metrics/tasks such as conversation event simulation, interruption response success rate, and inference latency.

**Claims And Evidence:**

Yes they mostly are but there is a problematic claim i describe below:

lines 68 & 134 => the paper's comparison to previous work (e.g., Moshi) may need revision for accuracy.

I think there is a potential issue in the paper’s positioning of prior work. Specifically, Moshi is inaccurately classified as following an encoder-decoder architecture when it actually employs a decoder-only model (Moshi is powered by Helium, a 7B parameter transformer-based language model designed for spoken dialogue. Helium follows a decoder-only structure, similar to GPT models, meaning it generates responses autoregressively without a separate encoder).

=>this is to me an important misunderstanding of a related work which weakens the paper positioning;

**Essential References Not Discussed:**

ok

**Experimental Designs Or Analyses:**

-The results in Table 1 need more detail—it's unclear what 0.1 or 0.9 represent in NTPP_0.1 or NTPP_0.9. Additionally, it's difficult to assess whether the differences shown in Table 1 (compared to dGSLM) and Figure 7 (compared to Moshi) are significant or meaningful. A clearer explanation   would help in understanding the impact.

-Inference latency is better => ok but where does it come from ? From the authors’ architecture or from leveraging  more recent KV-cache optimization techniques ?

-The most convincing exp to me  is the human evaluation, which measures turn-taking naturalness and content meaningfulness. However, more details on the evaluation process would be helpful—such as how listeners were recruited, their backgrounds, etc.

**Methods And Evaluation Criteria:**

Yes they are (although i have a few concerns expressed in the 'experiments & design' section)

**Other Comments Or Suggestions:**

no

**Other Strengths And Weaknesses:**

i covered them already in the previous sections

**Questions For Authors:**

section 5.8 ablation:
-can you elaborate more on what the two stage training approach is exactly (that seems to be better) => should be more detailed ?

**Relation To Broader Scientific Literature:**

It's not clear how original this work is compared to Moshi (Defossez, 2024). Their Next-Token-Pair Prediction (NTPP) method is new, but the idea of handling dual-channel speech in a single sequence using subsequences is not (done in Moshi technical report for instance).  Authors should more clearly explain what sets their approach apart.

**Theoretical Claims:**

The paper is mostly experimental, formalizations mostly recap previous work

---

> ### Author Rebuttal · Authors · 2025-04-01
>
> **Q1, Why we classify Moshi as a encoder-decoder model**
>
> Thanks for raising this important and insightful question. While Moshi utilizes the Helium LLM as its temporal transformer—a decoder-only architecture—the inclusion of the RQ-Transformer introduces a spatial transformer component, which deviates from a standard decoder-only structure for several reasons (see Figure 3 on page 13 and Figure 1 on page 7 in Moshi’s paper). First, the RQ-Transformer employs a spatial transformer that encodes input structure information before the autoregressive decoding process. This encoding step explicitly transforms the input data, making it functionally similar to an encoder stage. Second, instead of directly processing tokens in a purely decoder-based manner, the RQ-Transformer constructs a context vector through the spatial transformer. This context vector resembles the latent representations typically computed by encoders in encoder-decoder architectures. Overall, Moshi’s architecture can be viewed as “two decoder-only architectures connected via an encoder-decoder module.” Consequently, unlike a purely decoder-only approach like our NTPP, Moshi requires maintaining two separate KV caches for its two decoder-only transformers. Therefore, Moshi is not a standard decoder-only architecture; instead, we classify it as an encoder-decoder model from a dual-channel modeling perspective. We refer the reviewer to our response to Reviewer muk3 for further details on this matter.
>
> **Q2:The major weaknesses of Moshi architectures compared to our NTPP and previous methods**
>
> We refer this reviewer to read the Experiments Designs or Analyses Q1 response for the Reviewer muk3 for further details about this question.
>
> **Q3: Further Clarifications on Table 1**
>
> We followed the evaluation settings for quality and statistical analysis of generated dialogues used in Moshi[^1]. Specifically, we selected 1,000 random 10-second prompts from the Fisher dataset and utilized NTPP and other baseline models to generate dual-channel speech continuations. For each prompt of NTPP, we generated 32 continuations across three different temperature settings [0.1, 0.5, 0.9], as temperature significantly impacts the results.
>
> **Q4: Inference Latency Explanations**
>
> We employ Voice Activity Detection (VAD) as the latency measurement benchmark, calculating system response delays exclusively during effective speech token segments. This approach more accurately simulates real-world conversational user experience. Our experimental results demonstrate that the latency improvement in multi-round dialogue primarily stems from our novel Next-token-pair Prediction mechanism: By independently calculating the attention values of dual-channel tokens, our method achieves fine-grained time alignment. In contrast, Moshi’s strategy of directly summing dual-channel token embeddings risks introducing inter-channel information interference.
>
> **Q5: Further Clarifications on Human Evaluation Experiments.**
>
> We followed the evaluation settings of Moshi and conducted the evaluation study with 25 annotators who possess native-level English proficiency. These annotators were recruited through a combination of academic networks and online platforms, ensuring a diverse range of backgrounds and experiences. Each annotator was thoroughly briefed on the evaluation criteria and the objectives of the study. We adapted the Mean Opinion Score (MOS) protocol, utilizing a 5-point Likert scale, to assess two key aspects: the Naturalness (N-MOS) of turn-taking and the Meaningfulness (M-MOS) of dialogue content. This approach allowed us to gather quantitative data on the perceived quality of the generated dialogues.
>
> **Q6: More Representative Demos.**
>
> We provide more dialogue examples at [Demo Page](audio-3059.pages.dev), which illustrates the performance of our NTPP model across several key scenarios, including multi-turn dialogues, "Interruptions & Reflective Pause" evaluation, and speech continuation. The audio samples demonstrate how NTPP handles these different situations.
>
> **Q7: More Explanations on Two-Stage Training.**
>
> We describe our two-stage training process in Section 5.1 (Dataset). The first stage involves training a text-based LLM on single-channel speech token sequences. In the second stage, we introduce our newly proposed NTPP paradigm for dual-channel speech learning, building upon the SpeechLM obtained in the first stage. The ablation results in Figure 9 aim to support a simple claim: the first-stage training is essential before the second-stage NTPP training. We have further clarified this ablation study with a slightly modified figure [Figure 9](audio-3059.pages.dev/figure9). This updated figure includes three curves: Full Two-Stage NTPP (NTPP), NTPP without the first stage (NTPP w/o-1), and NTPP without the second stage (NTPP w/o-2).
>
> Please feel free to ask if you have any further question.

---

### Official Review · Reviewer_muk3 · 2025-03-12

**Overall Recommendation:** 3

**Summary:**

This paper proposes a next-token-pair prediction approach for modelling a dual-channel streamable Speech LM. The authors propose to use an autoregressive LM to model both speakers in a conversation, predicting token pairs from both channels at each timestep. The model is trained using a two-stage pipeline and compared against seminal works like dGSLM and Moshi.

**Claims And Evidence:**

The paper makes several claims regarding the advantages of their proposed architecture. However, there are some issues where the claims are not fully supported by evidence:

1. Encoder-decoder inefficiency (Lines 70-72): The authors claim that encoder-decoder models are inefficient and not scalable but provide no justification or supporting references. Given that Moshi, which (according to the authors) also follows an encoder-decoder structure, achieves the same goals as this paper, a more detailed explanation and citations are needed to support their claims.

2. (Lines 90-105): The authors list four advantages of their approach, but Moshi already exhibits three of these (points 2, 3, and 4) during pretraining. The differences between NTPP and Moshi should be more clearly articulated. Also, Moshi too uses decoder-only architecture as the inference is done by Helium LLM.

3.The performance of the RVQ-based tokenizer is not compared against Mimi or other streaming-compatible tokenizers, making it difficult to assess its effectiveness.

4. The performance improvements claimed in Table 1 and 2 cannot be properly verified due to insufficient details about baseline models (including the "cascaded" model) and evaluation methodology.

5. The paper highlights that its approach outperforms Moshi in inference latency as number of turn-taking increase, but since Moshi uses a Helium LLM with a smaller context window than Llama 3.1 8B, this might be an unfair comparison.

**Essential References Not Discussed:**

-

**Experimental Designs Or Analyses:**

1. The comparison against Moshi is unclear, especially given that the open-source Moshi checkpoint is fine-tuned on one speaker and always begins with a welcome message. The paper doesn't specify how this was handled while conducting evals.
2. The "one-stage" vs. "two-stage" ablation study in section 5.8 lacks clear definition. It's not clear to me what the one-stage approach is - details are missing from the paper.
3. Regarding Figure 7 - it's not clear who the "judge" is or the methodology used, making it difficult to assess the objectivity of these results.
4. Vocoder streaming compatibility is not addressed. How did the authors make the vocoder compatible for streaming?
5. Figure 6 is included but never discussed in the text and lacks a legend, making its purpose and meaning unclear.
6. Table 2 has limited comparisons. I would have also wanted to see comparison against Llama-Omni or SpeechGPT.

**Methods And Evaluation Criteria:**

The authors use relevant benchmarks like IPU, but some important details are missing:

1. The 14k-hour dataset composition is not clearly specified: Which three datasets were used? What is their language composition? Are they purely single-channel or multi-channel datasets?
2. The paper compares against Mistral-7B and Gemma-7B, but Qwen 2.5 would have been a more appropriate choice due to its stronger audio capabilities.
3. The paper does not use StoryCloze or ZeroSpeech Challenge. Other works like dGSLM and Moshi use these, so their absence weakens the evaluation.
4. The distinction between different NTPP variants (0.1, 0.5, and 0.9) in Table 1 is never explained, making it impossible to interpret these results meaningfully. There's also no mention of cascaded model anywhere in the main text.

Many more crucial details are missing from the paper which make it difficult to assess the validity of claims.

**Other Comments Or Suggestions:**

1. Lines 352-353:  Text says Appendix 1, but this seems to be missing (likely Appendix C.2). The reference should be fixed, and additional missing details should be included.
2. In Table 1, cascaded model has the lowest delta IPU of 1.3s yet the NTPP's values are highlighted.
3. Line 186, column 2. Typo: "modelling Figure 1" Wrong reference. Should be Figure 3.5.

**Other Strengths And Weaknesses:**

## Strengths:
- Proposed architecture looks simple to implement and can also work with new LLMs out-of-the-box.

## Weaknesses:

- Key implementation details are missing, making it difficult verify the evals and draw comparisons against Moshi.
- There are several references and clarity issues. Moreover, figure 6 is added but never discussed. [See "Questions for Authors" section below]
- I also have concerns around RVQ and Vocoder [See "Questions for Authors" section below]
- No standard benchmarks like StoryCloze or ZeroSpeech challenges were used for linguistic quality evaluation

**Questions For Authors:**

1. The RVQ implementation has ambiguity: it's unclear how Z_a and Z_b are handled in the RVQ case, specifically whether W_q is multiplied by d (calculated in equation 12) and how the model learns different embedding values for different speech tokens.
2. What are the three speech datasets that make up the 14,000 hours of training data? Please provide details about their language compositions and other characteristics.
3. What is the "cascaded model" referenced in Table 1, and how was it implemented? Also, please clarify what NTPP 0.1, 0.5, and 0.9 represent in this table.
4. How did you evaluate Moshi given that the open-source checkpoint is fine-tuned on one speaker (moshiko / moshika) and always begins with a welcome message?
5. How does your RVQ tokenizer compare to other streaming-compatible tokenizers like Mimi in terms of performance? What was the data used to train it?
6. Is your vocoder streaming-compatible? What are its encoding/decoding rates and how many frames of audio are synthesized in each time step?
7. Could the latency differences with Moshi be attributed to differences in context window size rather than architectural advantages of your approach?
8. What start tokens are used, and how exactly did you prepare the pretraining and finetuning data?
9. What does Figure 6 represent, and why is it not discussed in the main text?
10. Lines 293-294 say "To encode the relative positional information of tokens, all three leverage rotary positional encoding (Su et al., 2024a)." - Can authors elaborate more on this? Isn't equation 8, 9 and 12 being used for embeddings?

**Relation To Broader Scientific Literature:**

There has been a huge interest in speech LMs, especially ones that can process dual-channels like Moshi, GPT 4o, etc. These models have application in Voice Agents and hence, this paper too tries to make a contribution towards such models. I appreciate the authors' intention towards open-sourcing the codebase and checkpoints.

**Theoretical Claims:**

n/a

---

> ### Author Rebuttal · Authors · 2025-04-01
>
> # Claims and Evidence
> **Q1: The inefficiency of Encoder-decoder architectures**
>
> The efficiency of decoder-only model is supported by various literature. For example, FlashAttention (Dao et al., 2022) and Parallelized decoding (Kumar et al., 2020). They show how decoder-only models optimize memory and speed compared to encoder-decoder alternatives.
>
> **Q3: StoryCloze linguistic quality evaluation**
>
> | Model |  sStoryCloze$\uparrow$ |
> |---|---|
> |Spirit-LM| 61.0 |
> |Moshi| 60.9 |
> |NTPP| 61.4 |
>
> # Experiments Designs or Analyses
> **Q1: The comparison between Moshi, NTPP our other approaches.**
>
> We illustrate the comparison across different approaches through the following table:
> | Different Models | Speaker-Independent | Encoder-free | VAD-free | Single KVCache | End-to-End |
> |---|---|---|---|---|---|
> | dGSLM | $\checkmark$ |  | $\checkmark$ | | $\checkmark$|
> | LSLM |  |  | | $\checkmark$ |$\checkmark$|
> | Moshi |  |  | $\checkmark$ |  |$\checkmark$|
> | **NTPP(ours)** | $\checkmark$ | $\checkmark$ | $\checkmark$ | $\checkmark$ |$\checkmark$|
>
> In general, only our NTPP simultaneously possess the five important properties compared to other approaches. Moshi is not speaker-independent and not encoder-free. Additionally, it requires two KV caches.
> (1) This Moshi architecture is not “speaker-independent”. dGSLM indicates that the model should learn the joint distribution $p(s^{b},s^{a})$ instead of any conditional distribution $p(s^{b}|s^{a})$ or $p(s^{a}|s^{b})$. To satisfy this property, dGLSM adopts a two-tower transformer that follows the Siamese encoder-decoder architecture to estimate $p(s^{b},s^{a})$. Moshi and LSLM are only learning the conditional distribution $p(s^{b}|s^{a})$, which inherently has the generalization issue when we just simply switch speaker's roles. Also, this problem cannot be easily solved by training the paired data alternatively since learning $p(s^{b}|s^{a})$ and $p(s^{a}|s^{b})$ sequentially is not equivalent to learning $p(s^{b},s^{a})$ directly. (2) Moshi is not an encoder-free model since it adopts the RQ-transofmer architecture. Further clarifications are in the Reviewer Ubh3 Q1 response. (3) Moshi requires storing two separate KV caches for its two decoder-only transformers, reducing memory efficiency. In contrast, our NTPP, as a typical decoder-only architecture, requires only a single KV cache, making it more memory-efficient.
>
> **Q2: Two-Stage Ablation Studies**
>
> We refer this reviewer to read **Reviewer Ubh3** Q7.
>
> **Q3 & 4: Figure 7 judge and more clarifications on baselines and evaluation metrics in Table 1 & Table 2.**
>
> To save space, we refer this reviewer to read **Reviewer Ubh3** Q3.
>
> **Q6: Llama-Omni and SpeechGPT Evaluations.**
>
> Llama-Omni and SpeechGPT, as single-turn QA audio model, face incompatibilities with these real-time streaming multi-turn conversational benchmarks. Fisher and CANDOR require real-time handling of dynamic context shifts, interruptions (barge-ins), and mid-conversation pauses—capabilities inherently absent in VAD-dependent architectures.
>
> **Supplementary Material & Suggestions**
>
> We have updated the demo and code [Page](audio-3059.pages.dev). Upon acceptance, we will release the full code and model weight publicly and create a well-organized project page for it. The reported digit for the cascaded model should be 4.3 instead of 1.3 (a typo). We will promptly correct this in the revised version.
>
> # Questions
> **Q1-Q2: RVQ and Vocoder Implementations.**
>
> Regarding the RVQ tokenizer and Vocoder, we followed the training settings as specified in Soundstream (Neil et al. 2021) and HifiGAN (Jungil et al. 2020), respectively. Additionally, we have updated the training code with detailed training procedures.
>
> **Q3: Cascaded models and Table 1 Explanation**
>
> Since the performance of the Cascaded model relies on VAD, we use "Cascaded model" as a general term to represent all non-interactive models, including multi-modal approaches and LLM-based cascading models. This class models do not directly learn the generation of dual-channel speech, so we do not introduce them in details. We will add short descriptions in the revised version. For the interpretation of NTPP subscripts in Table 1, please refer the Q3 response for the Reviewer Ubh3.
>
> **Q4-Q6: Dataset details, Moshi Evaluation Details and Tokenizer comparison**:
>
> We put the corresponding content in [Page](audio-3059.pages.dev).
>
> **Q7: Inference Latency Analysis**
>
> To save space, we refer this reviewer to read Q4 response for the Reviewer Ubh3.
>
> **Q8: start tokens.**
> Instruction format:
> ```
> <bos><Model_0_0>...<Model_0_k><Human_0_0>...<Human_0_k><eos>
> ```
>
> **Q9: Figure 6 Discussions & Q10: Further explanations on positional encoding**
>
> We put Figure 6 discussions and further explanations on positional encoding in [Page](audio-3059.pages.dev).
>
> We hope the above responses address your concerns. If you have further questions, please feel free to ask.

---

> > ### Comment · Reviewer_muk3 · 2025-04-05
> >
> > Thank you for the responses to my concerns. While the authors have addressed several issues in their demo page (note that the link for demo page provided on this forum is incorrect - I had to use the link mentioned in the paper), there are still important points that need clarification:
> >
> > 1. **sStoryCloze Evaluation**: The authors' comparison appears to use Moshi's numbers after multi-stream instruct with synthetic voice variant from Table 7 of the Moshi paper. This is inappropriate for a fair comparison, as NTPP isn't fine-tuned on one synthetic voice. Comparing with Moshi's multi-stream variant (which achieves 62.7) would be more appropriate. This higher baseline score also then casts doubt on NTPP's claimed conversational capabilities. Please address this discrepancy.
> > 2. **Training Data Inconsistency:** The paper mentions using 14,000 hours of training data in stage 1, but the demo page states 140,000 hours. This 10x discrepancy is extremely misleading and hasn't yet been updated in the paper.
> > 3. **Figure 6 Presentation:** While I appreciate the addition of Figure 6 discussion on the demo page, this information should be included in the paper itself (or at minimum, in an appendix). The current state of the paper's presentation needs significant improvement to meet the conference's standards.
> > 4. **Latency Analysis:** Thank you for providing latency comparisons with similar context window sizes for Moshi and NTPP-Llama2. However, the numbers appear quite close. Please include standard deviations alongside the mean values for the 5 turn-taking conversations to provide a complete picture of the performance differences.
> > 5. **Tokenizer Evaluation:** The comparison table on the demo page lacks information about the evaluation methodology. What benchmark or dataset did the authors use to evaluate the "meaningfulness" and "naturalness" of these tokenizers? Without this context, it's difficult to assess the validity of the comparisons.
> > 6. **Moshi Evaluation Methodology:** The authors still haven't addressed how they evaluated Moshi given that the open-source checkpoint is fine-tuned on one speaker and always begins with a welcome message. Did they wait for this greeting to complete before starting the evaluation?
> >
> > Overall, I feel that both, the presentation quality and methodological clarity need significant improvement inorder to raise my score to atleast a weak accept.

---

> > > ### Author Response · Authors · 2025-04-05
> > >
> > > We greatly appreciate your further comments and questions, particularly your efforts in reading our demo page. We apologize for the confusion regarding the demo page link. The issue arises because the OpenReview platform automatically adds the "openreview.net/" prefix to the link. The correct link for the demo page is https://audio-3059.pages.dev. Here are our responses:
> > > 1. Thank you for raising this question. For a fair comparison, the **audio-only** result reported in Table 7 of the Moshi paper should be used, as NTPP does not include access to text alignment training data. This text alignment data is proprietary to the Moshi team and is not publicly available, which prevents us from training a text-based version of NTPP (i.e., by incorporating an additional text channel). The audio-only StoryCloze score in Moshi is **58.7**, which is notably lower than our NTPP score of **61.4**. We believe this performance gap could be further widened with large-scale single-channel pretraining comparable to Moshi’s setup. We initially reported the score from the synthetic voice variant because Moshi only publicly released the model weights for that version.
> > >
> > > | Model | sStoryCloze$\uparrow$ |
> > > |---|---|
> > > |Spirit-LM| 61.0 |
> > > |**Moshi (Audio only)**| **58.7** |
> > > |Moshi (Text and Audio, with synthetic data)| 60.9 |
> > > |**NTPP**| **61.4** |
> > >
> > > 2. The correct number should be 140K, as shown in the latter part of the demo page. We apologize for the inconsistency. We were aware of this typo; however, the paper cannot be updated at this stage. Rest assured, this will be corrected in the final version. We sincerely appreciate your careful reading and will ensure that all figures are accurate in the camera-ready submission.
> > >
> > > 3. Yes, we will replace the current Figure 6 with the updated, polished version on the demo page in the camera-ready submission. This rebuttal will be made publicly available and we will diligently uphold our promise.
> > >
> > > 4. We respectfully emphasize that NTPP-Llama2 reduces latency by 21.67% compared to Moshi, which we believe cannot be considered "quite close." Additionally, we have included the standard deviation alongside the mean values for five turn-taking conversations in our latency analysis, providing a more comprehensive view of the performance differences between Moshi and NTPP-Llama2.
> > >
> > > | Model | Audio response latency for 5 turn-taking(ms) | Standard Deviation |
> > > |---|---|---|
> > > | Moshi | 261.6 | 9.75 |
> > > | NTPP-Llama2 | 204.9 | 6.41 |
> > >
> > > 5. We would like to clarify that the "meaningfulness" and "naturalness" metrics are evaluated using the same settings as outlined in Table 2 of our paper. These metrics are based on the average results from the Fisher and CANDOR test sets. Furthermore, we have expanded our evaluation by incorporating turn-taking benchmarks to provide a more comprehensive comparison of the tokenizers. We train the NTPP model on Fisher dataset with different audio tokenizer. We split dataset by conversations into 6:2:2 for train, validation, and test respectively. We evaluated our model on the in-domain switchboard test set and additionally on 2 out-ofdomain (OOD) datasets: the Switchboard Corpus[1] and the Columbia Games Corpus[2] .
> > >
> > > |Audio Tokenizer | Meaningfulness $\uparrow$ | Naturalness $\uparrow$ | ROC-AUC of turn-taking label in Fisher $\uparrow$ | ROC-AUC of turn-taking label in Switchboard $\uparrow$ |  ROC-AUC of turn-taking label in Columbia Games  $\uparrow$ |
> > > |---|---|---|---|---|---|
> > > |Mimi| 4.05 | 4.28 | 83.22 | 84.38 | 82.25 |
> > > |Vanilla RVQ|3.95|4.15| 83.05 | 83.85 | 81.90 |
> > >
> > > **Audio Quality Metrics: Meaningfulness & Naturalness**
> > >
> > > **Interaction Response Accuracy: ROC-AUC of Turn-Taking Labels on two datasets Fisher & Switchboard**
> > >
> > > Our Vanilla RVQ model achieves competitive performance despite being pre-trained without any text data.  We believe this distinction underscores the potential of our Vanilla RVQ model in scenarios where text data may not be available or feasible to use.
> > >
> > > 6. Yes, we do wait for the greeting to complete before starting the evaluation. This ensures that the initial welcome message does not interfere with the latency measurements and allows for a more accurate assessment of Moshi's performance during turn-taking conversations.
> > >
> > > **End of Response:**
> > > We believe these improvements thoroughly address all the concerns raised and significantly enhance the overall quality of the paper. If so, we would greatly appreciate your consideration in updating your rating. Thank you once again for your valuable suggestions, thoughtful feedback, and especially your patience in reviewing the detailed responses.
> > >
> > > [1]: Godfrey, John J., Edward C. Holliman, and Jane McDaniel. "SWITCHBOARD: Telephone speech corpus for research and development." Acoustics, speech, and signal processing, ieee international conference on. Vol. 1. IEEE Computer Society, 1992.
> > >
> > > [2]: Gravano A, Hirschberg J. Turn-taking cues in task-oriented dialogue[J]. Computer Speech & Language, 2011, 25(3): 601-634.

---

### Official Review · Reviewer_eH4e · 2025-03-14

**Overall Recommendation:** 4

**Summary:**

The paper presents a method for improving the conversational capabilities of speech language models using dual-channel spoken dialogue learning. It introduces a Next-Token-Pair Prediction (NTPP) approach within a decoder-only transformer architecture, enabling the model to predict both speakers' next speech tokens simultaneously. The study evaluates this method using benchmarks and analyzes its performance in turn-taking dynamics, interruptions, and response timing, comparing with existing models such as Moshi and dGSLM.

## update after rebuttal
I have read the authors' reactions to my review. My recommendation was already very positive and I'm keeping it.

**Claims And Evidence:**

The claims are well supported by the evidence from the experiments carried out. The details of some of the evaluations are a bit sparse though (see below).

A minor problem is the claim that pretraining is done in a textless fashion. While defensible in a narrow sense, this is misleading as the starting point of the pre-training is a trained textual LLM.

**Essential References Not Discussed:**

None identified.

**Experimental Designs Or Analyses:**

The evaluation in section 5.5 is described only very briefly, lacking most details needed to understand its value. It's not clear what these automatically generated interactions sound like, or how exactly participants were asked to rate them.

**Methods And Evaluation Criteria:**

The evaluations are appropriate in general.

**Other Comments Or Suggestions:**

It may be useful to discuss how feasible it would be to extend this approach to more than two channels. Conversations with more than two participants are common, and people handle then easily, and thus thus capability would be useful to have in a dialog system.

**Other Strengths And Weaknesses:**

I don't understand what the two audio samples on the demo page are supposed to illustrate. They don't seem related to modeling dual channel audio in any obvious way.

**Questions For Authors:**

NA

**Relation To Broader Scientific Literature:**

The paper's main contribution is the introduction of a decoder only modeling for dual channel speech data via next token pair prediction.

**Theoretical Claims:**

NA

---

> ### Author Rebuttal · Authors · 2025-04-01
>
> Thank you for recognizing the value of our work. Here are our responses:
>
> **Q1: Textless Pretraining.**
>
> Thanks for your question. In our paper, "textless pretraining" specifically refers to the fact that neither the first-stage single-channel speech pretraining nor the second-stage dual-channel learning incorporates text alignment, unlike some existing SpeechLM models. We recognize the potential ambiguity and will provide a clearer explanation in the camera-ready version.
>
> **Q2: Section 5.5 Further Clarifications.**
>
> In our study, we aimed to assess the performance of existing audio models in real-world scenarios, particularly focusing on their ability to discern when to start and stop speaking. This is crucial for improving user experience, as current models often rely on Voice Activity Detection (VAD) to determine speech initiation, which can lead to interruptions whenever any sound is detected.
>
> We generated 200 thoughtful pauses (e.g., reflective pause or conversational hesitations like "Hmm... let me think...") and 200 interruptions (e.g., abrupt interjections: “But wait—[interrupted]”) using GPT-4o, ensuring contextual diversity. Audio was synthesized via ChatTTS with explicit silence annotations. We compared the performance of our model against two other models, Cascaded and Moshi, by measuring the proportion of instances where the VAD model correctly identified whether speech should occur.
>
> To ensure alignment between generated audio and labels, we employed human judge as the "gold standard". The closer this proportion was to human judgment, the better the model's performance. Figure 7 shows that our model is closer to human performance than Cascaded and Moshi.
>
> We have also added a demonstration of the corresponding audio effects on the [Demo Page](audio-3059.pages.dev).
>
> **Q3: About the two audio samples in demo page.**
>
> We provide more dialogue examples at [Demo Page](audio-3059.pages.dev), which illustrates the performance of our NTPP model across several key scenarios, including multi-turn dialogues, "Interruptions & Reflective Pause" evaluation, and speech continuation. The audio samples demonstrate how NTPP handles these different situations.
>
> Each audio sample on the demo page provides a clear representation of the output dynamics for each channel at any given moment. This setup allows users to observe how the model manages dual-channel audio, particularly in terms of when to initiate and cease speech. By listening to these samples, users can gain insights into the NTPP's ability to handle interruptions and reflective pauses effectively, showcasing its practical application in real-world interactions.
>
> **Q4：About Multi-channel system (more than channels)**
>
> Thank you for your suggestions. Expanding our approach to multi-channel conversation modeling is definitely on our roadmap, and our method can naturally be extended to this more complex setting. However, due to the lack of publicly available datasets, we are not yet fully prepared for this extension. Nonetheless, this remains a promising direction for our future work.

---

### Decision · Program_Chairs · 2025-05-01

**Decision:**

Accept (poster)

**Comment:**

This paper proposes a method to enhance the conversational abilities of speech language models through dual-channel spoken dialogue learning. The core innovation is the Next-Token-Pair Prediction (NTPP) approach, implemented within a decoder-only Transformer, which allows the model to simultaneously predict the next speech tokens for both speakers.

All reviewers agreed that the method offers valuable contributions to the spoken language modeling community. However, concerns were raised about its similarity to prior work, particularly Moshi. The authors addressed most of these concerns during the rebuttal, which is greatly appreciated. I encourage the authors to incorporate the additional results into the final version and to more clearly articulate how their approach differs from previous methods—especially those involving speech-text token mixing in Speech LMs and more exact definition of the encoder-decoder vs. decoder only architectures.